# Isolation, Identification, and Biocontrol Mechanisms of Endophytic *Burkholderia arboris* DHR18 from Rubber Tree against Red Root Rot Disease

**DOI:** 10.3390/microorganisms12091793

**Published:** 2024-08-29

**Authors:** Xiangjia Meng, Youhong Luo, Xinyang Zhao, Yongwei Fu, Lifang Zou, Haibin Cai, Yi Zhou, Min Tu

**Affiliations:** 1National Key Laboratory for Tropical Crop Breeding, Rubber Research Institute, Chinese Academy of Tropical Agricultural Sciences, Sanya 572024, China; 2023730089@yangtzeu.edu.cn (X.M.); 2021710770@yangtzeu.edu.cn (Y.L.); xyzhao1219@163.com (X.Z.); fu65227795@163.com (Y.F.); haibin_cai@163.com (H.C.); 2School of Agriculture, Yangtze University, Jingzhou 434000, China; 3Shanghai Collaborative Innovation Centre of Agri-Seeds, School of Agriculture and Biology, Shanghai Jiao Tong University, Shanghai 200240, China; zoulifang202018@sjtu.edu.cn; 4Sanya Research Institute, Chinese Academy of Tropical Agricultural Sciences, Sanya 572020, China

**Keywords:** *Burkholderia arboris*, rubber red root diseases, *Ganoderma pseudoferreum*, biological control

## Abstract

Red root rot disease is a devastating fungal disease of rubber trees caused by *Ganoderma pseudoferreum* (Wakef). Biocontrols using beneficial microorganisms are safe and sustainable. We isolated a DHR18 endophytic bacterium from a healthy rubber tree to obtain a new efficient antagonistic bacterium for red root rot disease affecting rubber trees and evaluated the mechanism of action involved using a double culture assay, genome annotation analysis, and the ethyl acetate extraction method. The results revealed that the DHR18 strain inhibits *G. pseudoferreum* growth and has broad-spectrum antifungal activity by secreting cell wall hydrolases (proteases and chitinases), indole-3-acetic acid, and siderophores. Furthermore, it fixes nitrogen and is involved in biofilm formation and phosphate solubilisation, improving disease resistance and tree growth. The results showed that the antifungal substances secreted by DHR18 are mainly lipopeptides. Simultaneously, DHR18 enhanced the rubber tree resistance to *G. pseudoferreum* by increasing the activities of defence enzymes superoxide dismutase, phenylalanine ammonia lyase, peroxidase, catalase, and polyphenol oxidase. The results indicate that *B. arboris* DHR18 has biocontrol potential and could be used as a candidate strain for the control of red root rot disease in rubber trees.

## 1. Introduction

The rubber tree (*Hevea brasiliensis*) is a perennial plant that produces 99% of the world’s natural rubber and has a wide range of industrial and commercial uses [1]. Red root rot is a soil-borne disease of rubber trees in China that significantly reduces the production period and yield, resulting in substantial losses to the rubber industry [2]. Red root rot disease is caused by *Ganoderma pseudoferreum*, which has a wide host range, long incubation period, and spreads rapidly [3,4]. Current control methods for red root rot disease include cultural and chemical treatments. Cultural control methods include mechanical root logging and the isolation of ditches. However, these measures are primarily related to land management and environmental improvements, which have significant limitations in disease prevention and control [5]. The prolonged application of chemical pesticides leads to pathogen resistance and affects the environment and human health [6,7]. Microbial inoculants are effective, environmentally friendly, sustainable, and have high reproductive rates [8,9].

Many studies have reported using beneficial microorganisms to control rubber tree diseases. For example, the filter paper method has been used to determine the *Bacillus velezensis* SF334 antibacterial activity against *Colletotrichum siamense* and *Colletotrichum australisinense* causing anthracnose in rubber trees and it revealed a 79.67% and 71.8% disease reduction, respectively [9]. The combined inoculation of *Enterobacter* sp. UPMSSB7, arbuscular mycorrhizal fungi (*Glomus mosseae*), and silicon considerably reduced the incidence and disease index of white root rot disease and significantly increased multiple growth parameters of rubber trees [8]. Chu et al. [10] found that the crude extract of lipopeptides from *B. velezensis* HN-2 enhanced the expression levels of several defence enzymes and genes in rubber tree leaves and inhibited the expression of the negative feedback regulatory gene HbLFG1 related to *Erysiphe quercicola* infection, indicating that HN-2 has significant biocontrol potential against powdery mildew in rubber trees. However, few studies have investigated the use of beneficial microorganisms to control red root rot disease in rubber trees.

Plant endophytes are plant-associated microorganisms that survive in the tissues of their hosts without harming the plants [11,12]. Plant endophytes produce various secondary metabolites [13]. Endophyte-derived bioactive compounds do not damage eukaryotic host systems; thus, they are less toxic to normal cells [14]. Previous studies suggested that microorganisms isolated from healthy plants are more likely to invade and colonise specific plant hosts without triggering defence mechanisms owing to evolutionary adaptation [15,16]. Endophytic bacteria can fix nitrogen and produce growth hormones to promote plant growth. They also produce antibacterial substances to inhibit and kill pathogenic microorganisms [17,18,19]. In addition, endophytic bacteria induce plant disease resistance by regulating the activity of plant defence enzymes and expression levels of defence genes [20,21]. For example, the endophytic bacteria *B. velezensis* D61-A [22], *Bacillus amyloliquefaciens* YZU-SG146 [23], and *Pseudomonas koreensis* B17-12 [24] from *Fraxinus hupehensis* significantly reduced rice sheath blight, cotton wilt disease, and postharvest diseases in tomatoes, respectively. Therefore, there could be abundant biocontrol bacterial resources and endophytic bacteria that have potential for use as biocontrol agents (BCAs).

Therefore, the present study seeks to screen and identify BCAs against *G. pseudoferreum*, evaluate the effects of the identified BCAs on rubber red root diseases under greenhouse conditions, the effect of the BCAs on the growth of rubber tree seedlings, and the antagonistic mechanisms involved. The objective is a green, effective, and sustainable method for the prevention and control of rubber tree red root disease.

## 2. Materials and Methods

### 2.1. Screening of Antagonistic Bacteria on Rubber Trees

The bacterial strain DHR18 was isolated from the roots of a healthy rubber tree at the Danzhou City Chinese Academy of Tropical Agricultural Sciences Test Ground, China, in June 2023. A dual-culture assay was used to determine the DHR18 antifungal activity against *G. pseudoferreum* (GenBank accession No: FJ374875). The DHR18 was inoculated in 50 mL of nutrient broth media (NB [Sigma-Aldrich, St. Louis, MO, USA]) and cultured in a shaker at 28 °C and shaken at 130 rpm for 2 days. The 5 mm-diameter plugs from *G. pseudoferreum* isolated from a red root rot diseased area at the Chinese Academy of Tropical Agricultural Sciences, Danzhou City, China were cultured for 5 days and transferred to the middle of flour dextrose agar (FDA) medium plates (30 g flour, 20 g glucose, 13 g agar, 1000 mL distilled water, and pH 7.0). Filter paper with a 5 mm diameter was soaked in the DHR18 bacterial culture and placed on three symmetrical points 25 mm from the centre of the FDA medium plates. Filter paper discs immersed in NB were used as the controls. Each treatment was repeated thrice, and all plates were cultured at 28 °C for 7 days. The diameters of pathogen colonies were measured, and inhibition rates were calculated as described by Chen et al. [25]. The inhibition rate was calculated using the following formula:[(R − r)/R] × 100%,
where ‘R’ is the radius of pathogen growth in control treatment, and ‘r’ is the radius of pathogen growth in treatment.

DHR18 was sent to the China Centre for Type Culture Collection (CCTCC) for identification and preservation in October 2023.

### 2.2. Strain DHR18 Antifungal Spectrum

To detect the broad-spectrum antifungal activity of the strain DHR18, it was tested against plant pathogenic fungi including *C. australisinense*, *C. siamense* (National Key Laboratory for Tropical Crop Breeding, Rubber Research Institute, Chinese Academy of Tropical Agricultural Sciences, Sanya City of Hainan province, China), *Botrytis cinerea*, *Magnaporthe oryzae*, *Colletotrichum phomoides*, and *Fusarium graminearum* (School of Agriculture and Biology, Shanghai Jiao Tong University, Shanghai, China). All fungi were cultured on potato dextrose agar (PDA) medium at 28 °C for 2 days, and the inhibition rates of strain DHR18 against six plant pathogenic fungi were detected as described in Section 2.1.

### 2.3. Effect of DHR18 on Rubber Red Root Diseases in the Greenhouse

The 3 L of medium containing 10 g Tryptone, 5 g yeast powder, 20 g glucose Kermel, 10 g Sucrose Kermel, 2 g Mg_2_SO_4_, 2 g KH_2_PO_4_, 0.2 g Vitamin b1, 15 g Agar, and 1000 mL distilled water, and at pH 7.0, was prepared, and 100 mL was divided into 30 triangular-250 mL mycelial plugs (5 mm in diameter) of the pathogen, which were cultured on a FDA medium for 3 days. Next, they were inoculated into a triangular bottle with a three-point symmetry 10 mm from the centre. All triangular vials were cultured at 28 °C, and plants were inoculated when mycelia filled the entire bottom of the medium (Appendix A). The DHR18 bacterial culture was prepared as described in Section 2.2 and diluted with the NB medium to 1 × 10^8^, 1 × 10^7^, and 1 × 10^6^ CFU/mL, denoted as P_3_, P_2_, and P_1_ treatments, respectively, and NB medium and tridemorph were used as negative and positive controls, respectively. The seedlings of the GTI rubber tree variety (30 cm long) were soaked in P_1_, P_2_, P_3_, and NB for 30 min and then transferred to triangular bottles containing *G. pseudoferreum,* cultured at 28 °C (L:D = 16:8) for 10 days, and, subsequently, the DHR18 control effect on rubber red root disease was determined.

There is currently no statistical method for determining the red root rot disease index in rubber trees. Therefore, we determined the DHR18 control effect by calculating the relative lesion area and gene expression of the pathogenic fungi. The relative lesion area was calculated using Assess 2.0 (American Phytopathological Society, APS). Control efficacy was calculated using the following formula:Control efficacy = (1 − relative lesion area in the control group/relative lesion area in the treatment group) × 100%.

We determined the effect of DHR18 on red root disease in rubber trees using real-time quantitative polymerase chain reaction (qPCR). The RNA Prep Pure Plant Kit (TIANGEN, Beijing, China) was used to extract total RNA from the mycelia of the root *G. pseudoferreum*. cDNA was synthesised using the ToloScript ALL-in-one reverse transcriptome EasyMix qPCR kit (TOLOBIO, Irvine, CA, USA). The cDNA was amplified using qPCR using the ACT (Forward: 5′-CATCGAGCACGGTATTGTCA-3′ and Reverse: 5′-TCTCGAACATGATTTGGGTC-3′) gene. The 18S rRNA (Forward: 5′-ACGAAGGTTAGGGGATCGAAA-3′ and Reverse: 5′-CGAGCGACACATAAGATTGAGG-3′) was used as the internal reference gene. The qPCR amplification reaction system included 2× Q3 SYBR qPCR Master Mix (10.0 μL [Cellagen Technology, San Diego, CA, USA]), DNA template (2.0 μL), forward and reverse primers (0.4 μL each), and ddH_2_O to make a final volume of 20.0 μL. The amplification procedure consisted of pre-denaturation at 95 °C for 3 min, followed by 35 cycles at 95 °C for 30 s, 55 °C for 30 s, and 72 °C for 1 min. The final extension step was performed at 70 °C for 8 min, followed by storage at 16 °C. Next, the cycle threshold value was derived, the correlation value was calculated using Excel 2010 (Microsoft Corporation, Redmond, WA, USA), and the relative expression of genes under different treatments was calculated according to the 2^−ΔΔCT^ Method [26].

### 2.4. Effect of DHR18 on Rubber Tree Seedling Growth in the Greenhouse

Strain DHR18 was inoculated in NB, and the culture was shaken at 130 rpm at 28 °C for 1 min to obtain a bacterial culture concentration of 1 × 10^8^ CFU/mL (OD_600_ = 1.0). The bacterial culture was centrifuged using a centrifuge (Thermo Fisher Scientific, Waltham, MA, USA) at 12,000 rpm for 30 min, and the bacterial pellets were collected and suspended in sterile distilled water to obtain a bacterial suspension. The bacterial suspension was diluted with Hoagland nutrient solution (Abiowell, Louisville, CO, USA) to 1 × 10^8^, 1 × 10^7^, and 1 × 10^6^ CFU/mL, denoted as T_3_, T_2_, and T_1_ treatments, respectively, and 100 mL of each treatment was added into 250 mL triangle bottles with rubber seedlings and cultured hydroponically in a greenhouse environment at 28 °C and light/day exposure time of 16: 8 h. Hoagland nutrient solution (Abiowell) without DHR18 was used as the blank control. The growth parameters of the rubber seedlings, including plant height, root length, fresh weight, dry weightand number of root system were measured 30 days after treatment, and the culture medium in the triangular bottle was changed after every10 days.

### 2.5. Analysis of the Characteristics of Endophytic Bacteria

#### 2.5.1. Siderophore Production Determination

The iron-producing carrier DHR18 was detected on chrome azure S (CAS) agar, considering the method described by Shin et al. [27]. A hole puncher with a 5 mm diameter was used to punch a hole 25 mm from the centre of the plate. DHR18 bacterial culture (100 μL) was injected into the hole, and the NB medium was used as the blank control. Each treatment was replicated thrice. All plates were placed in an incubator (Thermo Fisher Scientific) at 28 °C for 2 days, and the transparent circle around the colony was measured. Each treatment was replicated thrice.

#### 2.5.2. DHR18 Nitrogen Fixation Ability

A nitrogen-free agar medium plate was prepared using a method described by Liaqat et al. [28]. DHR18 colonies cultured on nutrient agar medium (Sigma-Aldrich) for 2 days were inoculated on nitrogen-free medium, and DH5α was inoculation and used as the control. Each treatment was replicated thrice. All plates were cultured in an incubator (Thermo Fisher Scientific) at 28 °C for 3 days, and the mycelial growth of the strains was observed.

#### 2.5.3. DHR18 IAA Production

DHR18 medium was inoculated into Yeast Malt Agar medium (Sigma-Aldrich) [9] and then oscillating cultured at 28 °C and 130 rpm for 72 h. The 1.5 mL bacterial culture was centrifuged using a centrifuge (Thermo Fisher Scientific) at 12,000 rpm for 10 min. Next, 0.5 mL supernatant was mixed with 1 mL Salkowski reagent (Loba Chemie, Mumbai, India), placed in a 1.5 mL Eppendorf tube (Eppendorf, Hamburg, Germany), and cultured at 25 °C in a dark room for 30 min. Next, 0.5 mL of the supernatant was mixed with 1 mL of Salkowski’s (1.5 mL 0.5 M FeCl_3_, 30 mL H_2_SO_4_, and 50 mL distilled water) reagent (Loba Chemie) in a 1.5 mL tube and cultured in the dark at 25 °C for 30 min. If the solution appeared pink, it was considered positive for indolic compounds [29].

#### 2.5.4. Detection of DHR18 Extracellular Enzymes

Skim milk (Sigma-Aldrich) [30], carboxymethyl cellulose (Sima-Aldrich) [31] and colloid chitin solid medium (Sigma-Aldrich) agars [32] were used to detect the DHR18 production of extracellular enzymes such as proteases, chitinases, and cellulases, respectively, and NB medium was used as a control. All plates were cultured at 28 °C for 3 days, and a clear circle surrounding the colonies was considered a positive enzymatic response, and its diameter was measured. Each treatment was repeated thrice.

#### 2.5.5. Determination of Phosphorus Dissolution Ability

The ability of DHR18 to dissolve phosphate was tested using the phosphate agar medium (National Botanical Research Institute, Uttar Pradesh, India) [33]. The DHR18 was inoculated as described in Section 2.5.1. NB medium was used as a control. The plates were cultured at 28 °C for 2 days, and clear circles around colonies indicated phosphate solubilisation by the strain.

#### 2.5.6. Detection of DHR18 Biofilm Formation

Biofilm formation by the DHR18 cells was determined using a colorimetric assay. The strain was inoculated on NB medium and cultured at 28 °C and shaken at 130 rpm using a shaker for 24 h. The bacterial suspension was diluted with NB medium to different concentrations (OD600 = 0.4, 0.6, and 0.8) and injected into glass test tubes with 4 mL of the bacterial suspension. Each treatment was repeated thrice with NB used as the control treatment and cultured at 28 °C for 48 h. A crystal violet staining solution (0.1%, *w*/*v*) was prepared and used to detect biofilm production. Next, 5 mm of crystal violet solution (Sigma-Aldrich) were added to each test tube, and the test tubes were left standing at 28 °C for 15 min. Next, the liquid was poured out and sterile water was used to wash free bacteria and excess dye solution. A purple ring mark on the tube wall indicated a biofilm had formed. Next, 5 mm of ethanol (95% *v*/*v*) was added to the test tubes, and they were slowly oscillated to dissolve the biofilm, and the absorbance of the solution was determined at 570 nm using a UV-visible spectrophotometer (PerkinElmer, Waltham, MA, USA). The initial absorbance was determined using a test tube containing NB as the control [34].

### 2.6. DHR18 Genomic Sequencing and Annotation

The strain DHR18 was inoculated in NB and cultured at 28 °C, 130 rpm min^−1^ oscillating to reach the logarithmic stage (OD_600_ was between 0.6 and 1.0). The DHR18 bacterial solution was centrifuged at 8000 rpm min^−1^ at 4 °C for 10 min, and the precipitation was collected and washed twice with 1 × phosphate buffer solution, and, finally, the supernatant was removed. The resulting bacterial sediment was frozen in liquid nitrogen for 15 min and sent to Benagen in Wuhan, Hubei Province, China, for whole-genome sequencing using Nanopore’s third-generation sequencing technology platform and Illumina’s second-generation sequencing technology platform (Illumina, San Diego, CA, USA).

After obtaining the raw data, connectors, short segments, and low-quality data were filtered. The filtered reads were assembled using Unicycler (version 0.5.0 [Illumina]), and high-accuracy Illumina data (Q30 > 85%) were assembled to obtain a high-quality bacterial genome skeleton (contig). The Nanopore data were then used to connect high-quality contigs to a finished graph.

The Genome BLAST Distance Phylogeny approach between DHR18 and homologous strains was calculated based on the DHR18 16S rDNA gene (GenBank ID: PP819542) and through the online JSpeciesWS platform (https://jspecies.ribohost.com/jspeciesws/#home accessed on 21 May 2024) to compare DHR18 with other genome average nucleotide identities (ANI).

The assembled genome was coded for gene prediction using Prokka (Version: 1.14.6) [35] software (Github, San Francisco, CA, USA). Prokka is a collection of gene element prediction tools called Prodigal, Aragorn, RNAmmer, and Infernal to predict coding genes, tRNA, rRNA, and miscRNA, respectively, and summarise and complete the preliminary annotation of various predicted gene elements. The assembled sequences were annotated using Gene Ontology (GO), Kyoto Encyclopedia of Genes and Genomes (KEGG), and Clusters of Orthologous Groups (COG). Secondary metabolite gene clusters were analysed and predicted using the antiSMASH online tool (https://antismash.secondarymetabolites.org/ accessed on 19 May 2024).

### 2.7. Antibacterial Activity of DHR18 Secondary Metabolites

#### 2.7.1. Antifungal Activity of Bacterial Culture Filtrate

Strain DHR18 was inoculated into NB (1 L), cultured at 28 °C, and shaken at 130 rpm using a rotary shaker (Thermo Fisher Scientific) for 7 days. The bacterial culture was centrifuged at 12,000 rpm at 4 °C for 30 min, and the supernatant was collected. The bacterial culture filtrate (BCF) of DHR18 was obtained by filtering the supernatant through a 0.22 μm-microporous filter. The BCF was mixed with PDA at 55 °C at concentrations of 5%, 10%, and 20% (*v*/*v*) to prepare plates. The PDA plate was mixed with an equal volume of NB as a control, and each treatment was replicated thrice. The centre of the PDA plate was inoculated with the mycelial plugs, and then the plate was cultured in an incubator (Thermo Fisher Scientific) at 28 °C until the control group mycelial covered the whole plate. The diameter of the mycelial growth diameter in each treatment was measured, and the inhibition rate was calculated as described in Section 2.1.

#### 2.7.2. Effect of Temperature on BCF Antagonistic Activity

The BCF of strain DHR18 was prepared as described in Section 2.7.1. The sterile fermentation solution was divided into six and placed in a room at 30 °C for 30 min, heated in a water bath at 60 °C for 30 min, heated in a water bath at 90 °C for 30 min, and placed in a high-temperature steriliser (Sigma-Aldrich) at 121 °C for 30 min. BCF treated at different temperatures were mixed with PDA at a concentration of 1:10 (*v*/*v*) to prepare plates. A PDA plate mixed with an equal volume of NB was used as a control, and each temperature treatment was replicated thrice. The *G. pseudoferreum* disk was inoculated at the centre of the PDA plate, and each plate was placed in an incubator set at 28 °C to make the control group. The diameter of the mycelial growth in each treatment was measured when the control group mycelia covered the entire plate. The inhibition rate was calculated as described in Section 2.1.

#### 2.7.3. Effect of Protease on the BCF Antagonistic Activity

The BCF of strain DHR18 was prepared as described in Section 2.7.1. Protease K was added to BCF to final concentrations of 0, 100, and 200 µg/mL. The mixture was cultured in an incubator (Thermo Fisher Scientific) at 37 °C for 1 h. BCF treated with different protease concentrations were mixed with PDA to prepare plates at 1:10 (*v*/*v*). BCF without protease and NB were used as controls. The diameter of the mycelial growth in each treatment was measured when the control group mycelia covered the entire plate. The inhibition rate was calculated as described in Section 2.1.

#### 2.7.4. Crude Extract Extraction and Detection of Antifungal Activity

The BCF of strain DHR18 was prepared as described in Section 2.7.1. Equal amounts of ethyl acetate were mixed with BCF, shaken in a separating funnel, and left to rest until fully stratified to collect the upper organic phase. The lower aqueous phase was continuously extracted by adding ethyl acetate thrice. Three organic compounds were combined, and ethyl acetate was removed using a rotary evaporation apparatus set at 65 °C; subsequently, the lipopeptide crude extract of strain DHR18 was obtained. The crude extracts were weighed and dissolved in 1 mL chromatograph-grade methanol and then added to 99 mL PDA medium at 55 °C. The contents of crude extracts in PDA were 500, 200, 100, and 50 μg/mL. The mycelial plugs were inoculated at the centre of the PDA plate, with three replicates for each treatment. Next, each plate was cultured in an incubator (Thermo Fisher Scientific) at 28 °C. When the control group mycelia covered the entire plate, the diameter of the mycelial growth in each treatment was measured, and the inhibition rate was calculated as described in Section 2.1.

### 2.8. Detection of the Defence-Related Enzyme Activities in Plants

The enzymatic activities related to the rubber tree root defence mechanism in strain DHR18 were examined as described in Section 2.4. The following treatments were used in the experiment: rubber tree roots were soaked in DHR18 bacterial culture at 1 × 10^8^ CFU/mL, 1 × 10^7^ CFU/mL, and 1 × 10^6^ CFU/mL for 30 min, denoted as Q_3_, Q_2_, and Q_1_, respectively; rubber tree roots were soaked in NB for 30 min for the control treatment. The aforementioned treatment was transferred to a conical flask containing *G. pseudoferreum* for processing, with conditions and methods similar to those described in Section 2.4. Rubber trees soaked in NB were transferred to a conical flask without *G. pseudoferreum* as a positive control [36]. At 1, 2, 3, 4, 5, and 6 days after treatment, the roots of the same plant were collected and subjected to rubber tree peroxidase (POD), phenylalanine ammonia-lyase (PAL), polyphenol oxidase (PPO), catalase (CAT), and superoxide dismutase (SOD) activity assays. Extraction of the defence enzyme solutions and determination of its activity were performed using a kit (Solarbio, Beijing, China). The experiment was repeated thrice.

### 2.9. Statistical Analysis

The data obtained in this study were analysed using a one-way analysis of variance using SPSS Version 23 software (SPSS Inc., Chicago, IL, USA). Duncan’s Multiple Range Test was used to compare the differences among the means of the treatment groups. Statistical significance was set at *p* < 0.05.

## 3. Results

### 3.1. Antagonistic Effect of DHR18 on G. pseudoferreum

The dual-culture assay results showed that the DHR18 inhibition of *G. pseudoferreum* was 83.89% (Figure 1). Strain DHR18 was sent to the CCTCC for identification and preservation, and the storage number of the strain was CCTCC No. M2024389. Strain DHR18 was identified as *Burkholderia*. sp. using 16S rRNA gene (GenBank: PP819542) identification in CCTCC (Appendix A).

### 3.2. Strain DHR18 Anti-Microbial Spectrum

Strain DHR18 showed inhibitory activity against all the tested pathogenic fungi (Figure 2 and Table 1). DHR18 exhibited the strongest inhibitory activity against *B. cinerea*, with inhibition rates exceeding 80%. DHR18 inhibitory effects against *C. siamense* and *F. graminearum* were comparatively weaker than those against *B. cinerea*, resulting in growth inhibition of 71.56% and 73.78%, respectively. The DHR18 inhibition rates against *M. oryzae*, *C. phomoides*, and *C. australisinense* were >75%.

### 3.3. Strain DHR18 Inhibitory Effect on Rubber Red Root Diseases in the Greenhouse

The DHR18 efficacy in controlling red root rot disease in rubber trees was tested by considering the relative lesion area and gene expression in *G. pseudoferreum* (Figure 3). The DHR18 concentration of 1 × 10^6^ (P_1_), 1 × 10^7^ (P_2_), and 1 × 10^8^ CFU/mL (P_3_) inhibited red root rot disease by 14.07, 23.13, and 62.11%, respectively, compared with that of the NB media. At DHR18 1 × 10^8^ CFU/mL, the inhibitory effect was equivalent to that of tridemorph (65.26%) (Table 2). At DHR18 concentrations of 1 × 10^6^, 1 × 10^7^, and 1 × 10^8^ CFU/mL, the efficacy against red root rot disease was 21.21, 38.40%, and 64.90%, respectively, compared with that of the NB media. At DHR18 1 × 10^8^ CFU/mL, the DHR18 efficacy was similar to that of tridemorph (70.99%) (Table 2). Furthermore, the two methods had consistent results.

### 3.4. Effect of DHR18 Effect on Rubber Seedling Growth

DHR18 promoted the growth of rubber tree seedlings (Figure 4 and Table 3). DHR18 significantly increased the number of rubber tree root systems (Table 3). T_3_, T_2_, and T_1_ DHR18 concentrations increased the number of rubber tree seedling roots by 142.39%, 47.09%, and 39.13%, respectively, compared with that of the control. A DHR18 concentration of 1 × 10^8^ CFU/mL (T_3_) significantly increased plant height, root length, dry weight, and fresh weight of rubber tree seedlings. DHR18 increased the plant height, root length, dry weight, and fresh weight of rubber tree seedlings by 11.67%, 19.77%, 13.78%, and 19.32%, respectively, compared with that of the control. A DHR18 concentration of 1 × 10^7^ CFU/mL (T_2_) significantly increased the fresh weight of rubber tree seedlings compared with that of the control. DHR18 concentration of 1 × 10^6^ CFU/mL (T_1_) had no significant effect on rubber tree seedlings.

### 3.5. Determination of Strain DHR18 Biological Characteristics

The medium inoculated with strain DHR18 exhibited protease hydrolysis, chitinase hydrolysis, phosphorus-solubilising transparent zones, and a yellow halo formed by iron ions (Figure 5A–C). DHR18 had normal growth on nitrogen-free plates (Figure 5F). Strain DHR18 can produce chitinase, protease, and siderophores with nitrogen fixation and phosphorus solubilising functions. DHR18 did not produce clear cellulose zones; therefore, it could not produce cellulase hydrolases (Figure 5D). DHR18 exhibited a pink colour reaction, indicating indole-acetic acid (IAA) production (Figure 5G). Strain DHR18 formed biofilms (Appendix A). At a concentration of OD570 = 0.8, strain DHR18 formed a significantly stronger biofilm than that at OD570 = 0.6 or 0.4 at *p* < 0.05.

### 3.6. Strain DHR18 Genome Sequencing and Annotation

After sequencing and assembly, the genome of DHR18 consisted of 7,386,926 bp with 66.75% GC content and 7450 protein-coding sequences), occupying 87.23% of the total chromosome length. The genome contained 73 tRNA, 5 5S rRNA, 6 16S rRNA, and 6 23S rRNA genes. (Appendix A). Furthermore, eight strains homologous to DHR18 were obtained using the Type Strain Genome Database (https://gctype.wdcm.org/ accessed on 26 March 2024). The DHR18 ANI value exceeded 96% (99.44%) that of *B. arboris* LMG24066, indicating that DHR18 is a *B. arboris* strain (Appendix A).

The COG annotation results showed that 6375 genes were annotated, relative to the COG database in DHR18, accounting for 85.57% of the predicted genes. These genes were annotated against 24 COG entries, of which 19 had >100 annotated genes. The most abundant genes were those involved in amino acid transport and metabolism (E), transcription (K), and general function prediction (R). DHR18 has genes encoding *β*-glucanase (COG2273), peptidoglycan hydrolases (COG3409), peptidoglycan/xylan/chitin deacetylase (COG0726), nitrogen fixation and metabolism regulation (COG5000). DHR18 also harboured multiple iron-carrier transport systems (COG3712, COG4114, COG4619, and COG0390) (Appendix A).

The GO annotation results showed that 5636 genes were annotated to the GO database in DHR18, accounting for 75.65% of predicted genes. Among them, 4159, 3014, and 3597 genes were involved in cellular processes, cellular components, and molecular functions, respectively. DHR18 contains genes related to colonisation facilitation, such as quorum sensing (GO: 0009372), biofilm formation (GO: 0043708), and spermidine biosynthesis (GO: 0008295, GO: 0004766 (Appendix A).

KEGG analysis results revealed that DHR18 had 2328 genes annotated relative to 23 pathways, accounting for 54.24% of the total number of genes, indicating that DHR18 has abundant substance metabolic pathways and can use various substances to meet its needs, making it adapt to the environment. Additionally, KEGG predicted that the DHR18 genes and products are involved in plant growth-promoting traits. For example, *trp*A, *trp*B, *trp*C, and aldh are involved in synthesising 1AA, and *nifQ* and *nifU* genes are related to nitrogen fixation (Appendix A).

AntiSMASH analysis showed 15 gene clusters related to secondary metabolite synthesis from DHR18. Ornibactin C8/C6/C4, pyochelin, and pyrrolnitrin had 100% similarity to known secondary metabolite synthesis gene clusters included in the database, with occidiofungin A exhibiting >85% similarity, indicating that DHR18 synthesises these secondary metabolites (Table 4 and Figure 6).

### 3.7. Effect of BCF on G. pseudoferreum Mycelial Growth

The DHR18 BCF significantly inhibited *G. pseudoferreum* growth (Figure 7A,B). The inhibition rates of *G. pseudoferreum* reached 71.76, 78.82%, and 84.71% at BCF concentrations of 2.5%, 5%, and 10%, respectively. There was no significant difference in the inhibition rates of BCF treated with different concentrations of Proteinase K (0, 100, and 200 μg/mL) against *G. pseudoferreum* (Figure 7C,D). There was no significant difference in the inhibition rates of BCF treated at different temperatures (30, 60, 90, and 121 °C) against *G. pseudoferreum* (Appendix A). Therefore, proteinase K and temperature do not affect the BCF antifungal activity. The antibacterial secondary metabolites in BCF were primarily thermostable lipopeptide substances.

### 3.8. Detection of Antifungal Activity of Crude Extract

Lipopeptide secondary metabolites from the DHR18 BCF were extracted using the ethyl acetate extraction method. The findings revealed that the lipopeptide-active substances of the DHR18 strain exhibited robust antifungal activity. The inhibition rates at 50, 100, 200, and 400 μg/mL concentrations were 28.69, 56.23, 65.23, 78.66, and 84.89%, respectively (Figure 8).

### 3.9. Effects of DHR18 on the Activity of Rubber Defence-Related Enzymes

Changes in the aPOD, PPO, PAL, CAT, and SOD activities in the roots of rubber tree seedlings after DHR18 and *G. pseudoferreum* treatments are presented in Figure 9. The enzymatic activities of the five enzymes treated only with pathogenic fungi (disease control) first increased and then decreased, with SOD and PAL, PPO and POD, and CAT activities peaking 3, 4, and 5 days after treatment, respectively. The activities of these five enzymes increased with increasing DHR18 concentrations. After DHR18 treatment, peak values of PPO and POD activities were observed at 2 and 4 days after treatment, respectively, and maximum values of 78.33 and 16.23 × 10^3^ U·g^−1^·min^−1^ FW were reached 4 days after treatment, which were 0.53- and 0.41-fold higher than those in the diseased control, respectively. The maximum PAL and SOD values of 86.36 and 742.59 U·g^−1^·min^−1^ FW were reached after 3 days of treatment, which was 0.61- and 0.29-fold higher than those of the diseased control, respectively. Peak values of CAT activity were observed 3 and 5 days after treatment, respectively, and reached a maximum value of 681.47 U·g^−1^·min^−1^ FW 3 days after treatment, which was 1.44-fold higher than that in the diseased control.

## 4. Discussion

Gram-negative *Burkholderia* can enhance plant growth through nitrogen fixation and phytohormone production. In addition, they can produce various metabolites with antibacterial activity, which can inhibit the proliferation of plant pathogenic fungi and bacteria [37,38]. Currently, some *Burkholderia* strains, such as *Burkholderia stabilis* EB159, *B. vietnamiensis* C12, and *Burkholderia* sp. SSG has been reported to have potential biocontrol activity against crop diseases and the capacity to promote crop growth [17,39,40]. However, there are no reports on the control of rubber tree red root disease using *Burkholderia* sp. In the present study, *B. arboris* DHR18 was isolated and identified. It had good preventive effects against rubber tree red root rot disease caused by *Ganoderma pseudoferreum*. The antagonistic mechanisms of DHR18 against *G. pseudoferreum* were investigated through whole-genome sequencing and siderophore and IAA analyses, in addition to enzyme activity assays. To the best of our knowledge, this is the first report of *B. arboris* strain as a potential biocontrol agent for rubber tree red root rot disease caused by *G. pseudoferreum*.

We isolated and identified a *B. arboris* strain, DHR18, effective against rubber tree red root disease caused by *G. pseudoferreum* and other plant pathogens, indicating that DHR18 possesses broad-spectrum antifungal activity, consistent with the characteristics of most bacteria used as biocontrol agents [41]. In the present study, the control effect of DHR18 against rubber tree red root disease and its growth-promoting effect on rubber tree seedlings was investigated by inoculation with different concentrations of DHR18 culture. We observed that the control effect became more pronounced as the inoculation concentration increased. When the concentration of DHR18 was 10^8^ CFU/mL, its control effect on rubber tree red root disease was not significantly different from that of the pesticide tridemorph, potentially due to the limitations of using a single strain. Therefore, in subsequent studies, the control effect of strain DHR18 could be enhanced through strain combination or co-inoculation with organic fertilizers [42,43]. In the growth promotion experiment, only when the concentration of DHR18 was 10^8^ CFU/mL could it significantly increase various growth indicators of rubber tree seedlings. Thus, DHR18 can only exert its growth-promotion effect at a certain concentration.

A previous study on the antifungal mechanism of DHR18 against *G. pseudoferreum* demonstrated that the strain DHR18 produces hydrolases (protease and chitinase) but not cellulases. Chitin and proteins are important structural components in the cell walls of plant pathogenic fungi and are essential for their survival and spread [44]. Therefore, strain DHR18 may inhibit the growth of pathogenic bacteria by secreting proteases and chitinases that deform the *G. pseudoferreum* hyphae. DHR18 fixes nitrogen, produces IAA and siderophores, and solubilises phosphorus, characteristics consistent with those for plant growth-promoting bacteria [19,38,45]. Siderophores secreted by beneficial bacteria promote plant growth under iron deficiency [46,47]. Previous studies have revealed that the strain DHR18 produces IAA and siderophores, fixes nitrogen, and solubilizes phosphorus, promoting rubber tree growth. The colonization potential of biocontrol bacteria is correlated with their ability to form biofilms [48,49]. Notably, strain DHR18 exhibited the ability to produce biofilms, indicating its colonisation ability.

In the present study, secondary metabolites produced by DHR18 had significant antifungal activity against *G. pseudoferreum*, even after treatment with high temperatures and proteinase K, consistent with the results obtained with *B. pyrrocinia* S17-377 and inconsistent with those obtained with *B. velezensis* D61-A [22,34]. Therefore, the antifungal active products produced by DHR18 and S17-377 were thermostable lipopeptides but not proteins. Prediction of gene clusters for synthetic secondary metabolites indicated that DHR18 may produce lipopeptide substances such as ornibactin, pyochelin, occidiofungin, and pyrrolnitrin. Simonetti et al. [50] and Jia et al. [51] conducted genetic mutation experiments and revealed that the antifungal properties of *B. amrifaria* T16 and *Burkholderia* sp. MS455 primarily depend on occidiofungin production. The produced siderophores, oribactin and pyochelin, facilitate iron uptake during plant growth in environments with low available iron [52,53,54,55]. Pyrrolnitrin was isolated and purified from *B. pyrrocinia* 2327T (DSM 10685T) [56] and developed as an agricultural fungicide against soil-borne fungal pathogens because of its broad-spectrum fungicidal activity [57]. Previous studies revealed that occidiofungin A, Pyrrolnitrin, and oribactin and pyochelin are active antifungal substances and siderophores produced by DHR18, respectively. Additionally, gene cluster prediction indicated that DHR18 could synthesise various unknown substances with potential antifungal activity.

Plant defence-related enzymes can eliminate reactive oxygen species in plants and trigger salicylic acid (SA), jasmonic acid (JA), and ethylene (ET) pathways to protect plants against pathogen infection [58,59,60,61,62]. Inoculation with DHR18 increased the activities of PAL, PPO, POD, CAT, and SOD activities in rubber trees in vivo. Similarly, the *Paenibacillus polymyxa* SC09-21 triggered the defence response of pepper against *Phytophthora capsici* by enhancing PPO, SOD, and PAL activities [61]; *B. altitudinis* GTS-16 induced the optimal synthesis of PAL and activated SOD during *Rhizoctonia solani* infection in rice [45]. DHR18 induces the premature expression of the defence enzymes in rubber trees and enhances their activities. Therefore, DHR18 increased rubber tree resistance to red root rot disease by increasing the activity of defence enzymes in rubber tree seedlings, indicating that the biological control activity of DHR18 against *G. pseudoferreum* may be related to various mechanisms and their synergies. Further studies are needed to identify other biocontrol mechanisms involved in the anti-microbial activity of DHR18 and the components of its antifungal compounds.

## 5. Conclusions

The bacterial strain DHR18 was isolated from the roots of rubber tree seedlings and exhibited strong inhibition of *G. pseudoferreum* growth, which causes rubber tree red root disease. The DHR18 sequence was identical to that of *B. arboris*. The results of a dual-culture assay greenhouse experiment revealed that strain DHR18 had significant inhibitory effects on *G. pseudoferreum*. Strain DHR18 produces hydrolytic enzymes and active substances that inhibit *G. pseudoferreum* growth and promote the growth of rubber trees through IAA and siderophore production and solubilising phosphorus. Moreover, strain DHR18 increases the activity of defence enzymes in rubber trees, inducing disease resistance. Thus, DHR18 has the potential for the prevention and treatment of rubber tree red root disease.

## Figures and Tables

**Figure 1 microorganisms-12-01793-f001:**
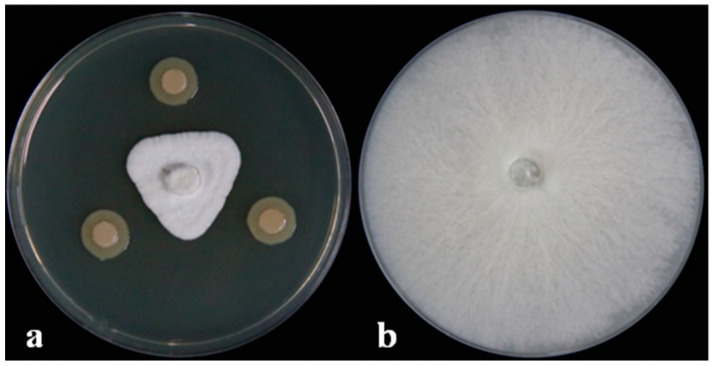
DHR18 antifungal activity against *Ganoderma pseudoferreum* (Inhibition rate:83.89%). Bacterial cultures (**a**) and sterile water (**b**) were inoculated on the three-point symmetry of a fungal plug.

**Figure 2 microorganisms-12-01793-f002:**
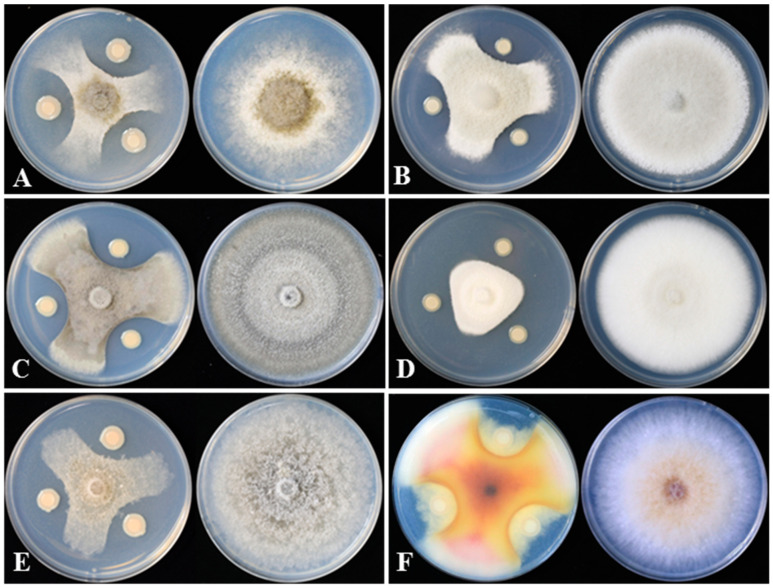
DHR18 inhibitory effects on six plant pathogenic fungi cultured on potato dextrose agar. (**A**) *Botrytis cinerea*, (**B**) *Colletotrichum siamense*, (**C**) *Colletotrichum phomoides*, (**D**) *Colletotrichum australisinense*, (**E**) *Magnaporthe oryzae*, and (**F**) *Fusarium graminearum*.

**Figure 3 microorganisms-12-01793-f003:**
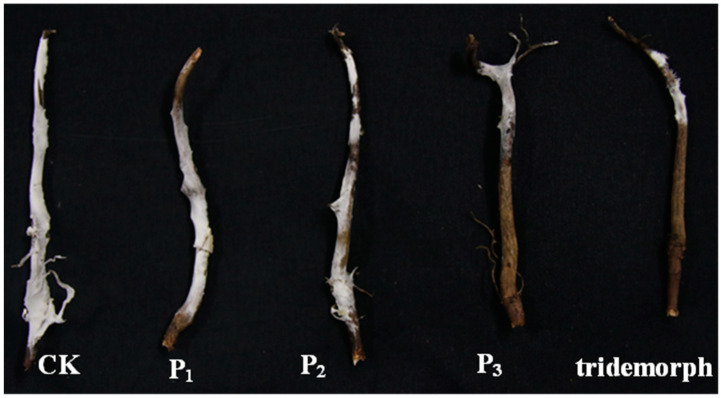
Roots of rubber tree seedlings infected with *Ganoderma Pseudoferreum* at different DHR18 concentrations under greenhouse conditions. P_3_: 10^8^ CFU/mL DHR18; P_2_: 10^7^ CFU/mL DHR18; P_1_: 10^6^ CFU/mL DHR18; CK: nutrient broth media.

**Figure 4 microorganisms-12-01793-f004:**
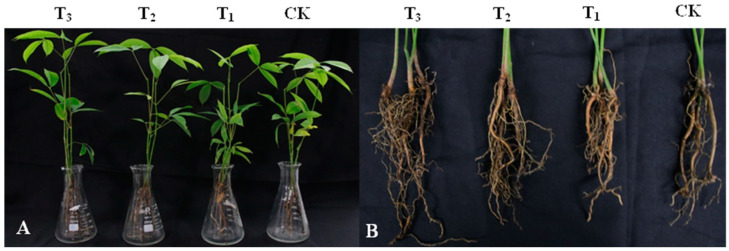
Effect of DHR18 on the rubber tree seedling growth. (**A**) Above-ground parts of rubber tree seedlings. (**B**) Underground parts of rubber tree seedlings. T_3_: 10^8^ CFU/mL oDHR18; T_2_: 10^7^ CFU/mL DHR18; T_1_: 10^6^ CFU/mL DHR18; CK: sterile water.

**Figure 5 microorganisms-12-01793-f005:**
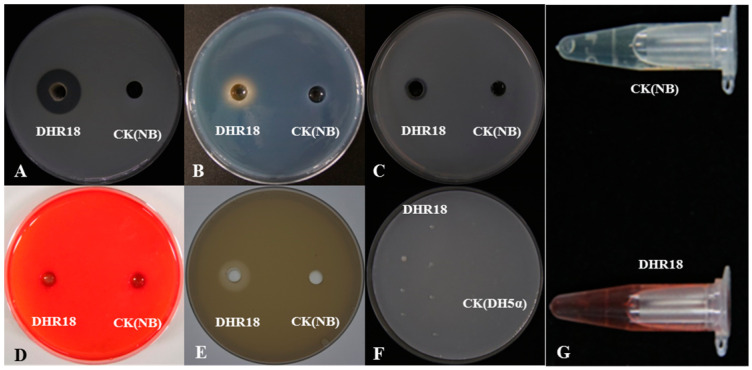
Detection of cell wall-degrading enzymes and siderophore: (**A**) protease, (**B**) siderophore, (**C**) chitinase, (**D**) cellulase, (**E**) phosphate solubilization, (**F**) nitrogen fixation, and (**G**) indole-acetic acid production. NB—nutrient broth media, CK—control.

**Figure 6 microorganisms-12-01793-f006:**
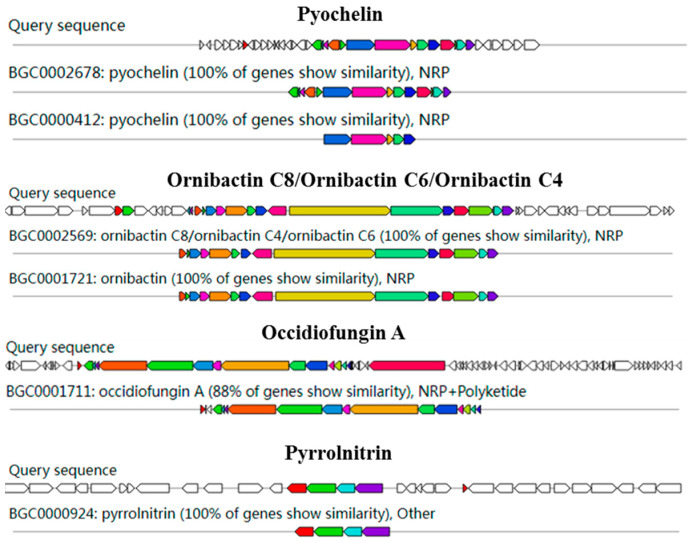
Genome annotation of DHR18 and prediction of biosynthesis gene clusters (BGC). Genomic information and chemical structure of BGCs compared to those of known BGCs.

**Figure 7 microorganisms-12-01793-f007:**
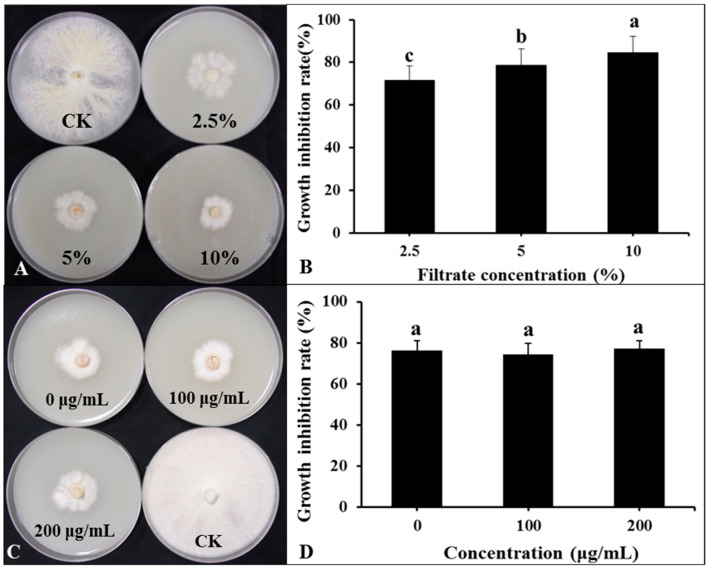
Effect of bacterial culture filtrate (BCF) amended to molten agar at different concentrations on mycelial growth of *Ganoderma pseudoferreum* after 7 days of incubation at 28 °C (**A**,**B**). Antibacterial activity of BCF (5%, *v*:*v*) treated with different concentrations of protease K against *G. pseudoferreum* after 7 days of incubation at 28 °C (**C**,**D**). Bars indicate the standard error of the mean. Columns marked with the same letter are not significantly different at *p* ≤ 0.05, considering Duncan’s Multiple Range Test (**B**,**D**).

**Figure 8 microorganisms-12-01793-f008:**
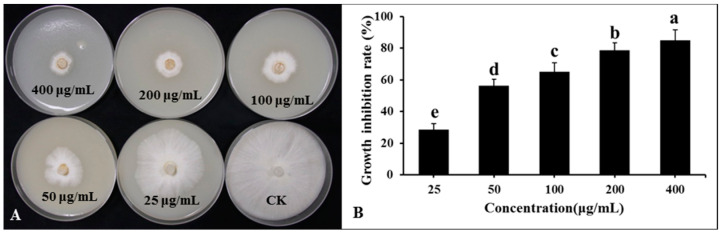
Inhibitory effect of strain DHR18 crude extract on *Ganoderma. Pseudoferreum* (**A**,**B**). Bars indicate the standard error of the mean. Columns marked with the same letter are not significantly different at *p* ≤ 0.05 considering Duncan’s Multiple Range Test (**B**).

**Figure 9 microorganisms-12-01793-f009:**
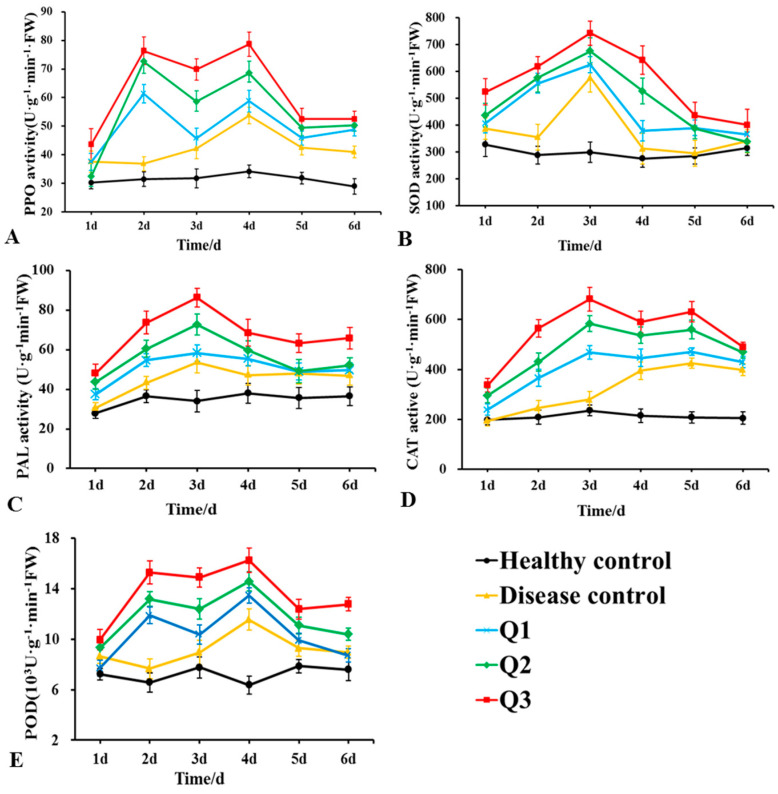
Effects of different concentrations of DHR18 on defensive enzyme activities in rubber tree seedlings at 1, 2, 3, 4, 5, and 6 days after treatment. (**A**) Effects of different treatments on polyphenol oxidase a, (**B**) superoxide dismutase, (**C**) phenylalanine ammonia lyase, (**D**) catalase activity of CAT, and (**E**) peroxidase activities. Q_1_: 1 × 10^6^ CFU/mL DHR18 bacterial culture treatment; Q_2_: 1 × 10^7^ CFU/mL DHR18 bacterial culture treatment; Q_3_: 1 × 10^8^ CFU/mL DHR18 bacterial culture treatment; Healthy Control: NB treatment; Diseased Control: Pathogenic fungi treatment.

**Table 1 microorganisms-12-01793-t001:** DHR18 anti-microbial activities against nine plant pathogens.

Pathogenic Fungi	Antibacterial Bandwidth (mm)	Inhibition Rate (%)
*Botrytis cinerea*	14.12 ± 1.15	82.22 ± 2.78 a
*Magnaporthe oryzae*	12.74 ± 0.80	78.89 ± 1.93 b
*Colletotrichum siamense*	9.70 ± 1.37	71.56 ± 3.31 d
*Colletotrichum phomoides*	12.10 ± 1.54	77.33 ± 3.71 b
*Colletotrichum australisinense*	11.45 ± 1.08	75.78 ± 2.61 bc
*Fusarium graminearum*	10.62 ± 1.86	73.78 ± 4.48 cd

The inhibition ratio was calculated as (R − r)/R × 100%, where ‘R’ is the growth distance of the hyphal plug facing each paper disk culture in control plates, and ‘r’ is that in the treatment plates. Values are presented as mean ± standard deviation of three replicates. Means that are followed by the same letter are not significantly different at *p* ≤ 0.05 considering Duncan’s Multiple Range Test.

**Table 2 microorganisms-12-01793-t002:** DHR18 efficacy against rubber red root rot diseases in the greenhouse.

Treatment	Relative Lesion Area (%)	Control Efficacy (%)	Relative Expression	Control Efficacy (%)
Tridemorph	32.47 ± 3.72	65.26 a	10.26 ± 1.23	70.99 a
DHR18 (Q_3_)	35.42 ± 6.21	62.11 a	12.41 ± 1.78	64.90 a
DHR18 (Q_2_)	71.35 ± 7.31	23.13 b	21.78 ± 2.74	38.40 b
DHR18 (Q_1_)	80.33 ± 8.24	14.07 c	27.86 ± 2.55	21.21 c
CK	93.48 ± 12.41	-	35.36 ± 3.21	

Q_1_: 10^6^ CFU/mL. Q_2_: 10^7^ CFU/mL. Q_3_: 10^8^ CFU/mL. Numerical values represent the mean ± standard deviation of the triplicate experiments. Means were tested using Duncan’s multiple range test using SPSS version 17.0 software. Means followed by the same letter within the same column are not significantly different (*p* ≤ 0.05).

**Table 3 microorganisms-12-01793-t003:** Effect of DHR18 on rubber tree seedlings growth in the greenhouse.

Growth Parameter	T_3_	T_2_	T_1_	CK
Plant height (mm)	34.27 ± 2.31 a	33.45 ± 1.98 ab	32.78 ± 1.06 ab	30.69 ± 1.36 b
Root length (mm)	13.45 ± 1.25 a	11.78 ± 1.13 b	11.36 ± 0.87 b	11.23 ± 0.69 b
Fresh weight (g)	5.78 ± 0.45 a	5.41 ± 0.32 a	5.34 ± 0.64 ab	5.08 ± 0.47 b
Dry weight (g)	2.47 ± 0.14 a	2.21 ± 0.11 ab	2.12 ± 0.14 ab	2.07 ± 0.21 b
Number of root system	22.3 ± 3.4 a	17.6 ± 2.1 b	11.8 ± 1.7 c	9.2 ± 1.6 d

T_1_: 10^6^ CFU/mL. T_2_: 10^7^ CFU/mL. T_3_: 10^8^ CFU/mL. Numerical values represent the mean ± standard deviation of the triplicate experiments. Means were tested using Duncan’s multiple range test using SPSS version 17.0. Means followed by the same letter within the same row are not significantly different (*p* < 0.05).

**Table 4 microorganisms-12-01793-t004:** Prediction of biosynthesis gene clusters (BCGs) in strain DHR18 genome.

Type	Start	End	Similar Cluster	Similarity
NRP-metallophore, NRPS	1,737,336	1,802,333	Ornibactin C8/C6/C4	100%
NRP-metallophore, NRPS	106,666	160,518	Pyochelin	100%
Other	1,084,700	1,125,785	Pyrrolnitrin	100%
NRPS, T1PKS, betalactone	539,589	675,375	Occidiofungin A	88%
NRPS, transAT-PKS	1,243,460	1,264,524	N-acyloxyacyl Glutamine	50%
Terpene	761,188	808,822	Capsular polysaccharide	16%
phosphonate	1,413,306	1,446,017	Dehydrofosmidomycin	15%
Arylpolyene	3,128,937	3,170,148	APE Vf	10%
acyl_amino_acids	3,144,169	3,205,188	Pf-5 pyoverdine	1%
NRPS-like, NRPS, T1PKS	1,849,817	1,902,941	-	-
Terpene	2,002,539	2,023,564	-	-
Hserlactone	523,186	543,794	-	-
NRPS-like, hydrogen-cyanide	2,038,092	2,091,743	-	-

NRP—non-ribosomal peptides, PKS—Polyketide synthases.

## Data Availability

The original contributions presented in this study are included in the article/Appendix A, and further enquiries can be directed to the corresponding authors. The DHR18 genome data presented in this study are available in the National Centre for Biotechnology Information (NCBI) database (GenBank number: GCA_037996665.1 at the NCBI database.

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
