# Peer review of "Isolation, Identification, and Biocontrol Mechanisms of Endophytic Burkholderia arboris DHR18 from Rubber Tree against Red Root Rot Disease"

_microorganisms, 2024, doi:10.3390/microorganisms12091793_

Round 1
Reviewer 1 Report
Comments and Suggestions for Authors
Microorganisms-3114724
Authors
The research is very complete and structured according to the objectives shown.
Minor comments
1.- In general, mention that a completely randomized experimental design was used in each experiment in section 2. materials and methods
2.- in lines 487 and 488 about the enzymatic activity of SOD and PAL, mention how many days there is a decrease
3.- In line 501 delete day 8, no appears in figure 9.
Author Response
Comments 1: In general, mention that a completely randomized experimental design was used in each experiment in section 2. materials and methods.
Response 1: We have revised the section 2, and I hope you can review it again.
Comments 2: in lines 487 and 488 about the enzymatic activity of SOD and PAL, mention how many days there is a decrease
Response 2: We have modified the problem in the manuscript, and I hope you can review it again (Line 485).
Comments 3: In line 501 delete day 8, no appears in figure 9.
Response 3: We have modified the problem in the manuscript, and I hope you can review it again (Line 499).

Reviewer 2 Report
Comments and Suggestions for Authors
Dear Editor
Many thanks for considering me as a potential reviewer for the article entitled "Isolation, identification and biocontrol mechanisms of endophytic Burkholderia arboris DHR18 from rubber tree against red root rot disease". No doubt the article is well-structured, presented and well-written. However, I have several observations, including which I believe should be taken into consideration before proceeding further.
My observations are as follow.
My observations
· Abstract; a) overall the abstract is fine, however, it can be a bit shorter, b) all scientific names should be italic.
· Introduction; a) Line-46, please add some statistics (which seriously shortens the production life and rubber yield of rubber trees, and brings huge losses to rubber industry) b), line-59, where is citation? (Currently, there have been many reports on the use of beneficial microorganisms to 59 control rubber tree diseases).
· Material and method; a), Line-201 remove (Guo et al., 2016), b), P≤0.05, p should be italic, c), confusing line-191 (DHR18 medium was inoculated in YM medium [9] and then oscillating incubated at 28°C and 130 rpm min-1 for 72h).
· Results and Discussion; I think it’s much refined, well-written and presented.
· References; Fine!
Minor comments
· All scientific names should be italic, throughout the manuscript,
· P≤0.05, p should be italic, do the said, throughout the manuscript,
Comments on the Quality of English Language
Dear Editor/Authors,
Thanks for the consideration!
The article is fine, however, I am not well qualified with English language i.e a number of issues detected and also some citations are wrongly inserted.
I would like to suggest this article for extensive English editng by a professional/native speaker and/or by a senior Prof. in the field.
Thanks!
Author Response
Comments 1: Abstract; a) overall the abstract is fine, however, it can be a bit shorter, b) all scientific names should be italic.
Response 1: Abstract; a) we have appropriately reduced the abstract of the manuscript, b) all scientific names have been modified. I hope you can review it again.
Comments 2: Introduction; a) Line-46, please add some statistics (which seriously shortens the production life and rubber yield of rubber trees, and brings huge losses to rubber industry) b), line-59, where is citation? (Currently, there have been many reports on the use of beneficial microorganisms to control rubber tree diseases).
Response 2: Introduction; a) We have modified the problem in the manuscript, and I hope you can review it again (Line-38). b) In response to this issue, we have provided illustrative examples, with citations [8], [9], and [10]all being cited (Line-48-57).
Comments 3: Material and method; a), Line-201 remove (Guo et al., 2016), b), P≤0.05, p should be italic, c), confusing line-191 (DHR18 medium was inoculated in YM medium [9] and then oscillating incubated at 28°C and 130 rpm min-1 for 72h).
Response 3: Material and method; a), We have removed (Guo et al., 2016) (Line-199), b), "P≤0.05" has been italicized throughout the manuscript, c), We have modified the problem in the manuscript (DHR18 medium was inoculated into Yeast Malt Agar medium (Sigma-Aldrich) and then oscillating cultured at 28 ℃ and 130 rpm for 72 h, Line-188)

Reviewer 3 Report
Comments and Suggestions for Authors
My comments can be found in the attached MS.

Comments on the Quality of English LanguageMy comments on English can also be found in the attached MS.
Author Response
Comments 1: Please provide information on other previously used biocontrol agents against Ganoderma pseudoferreum
Response 1: We apologize for this issue, but it seems that there are currently no reports on the use of biocontrol agents to control Ganoderma pseudoferreum.
Comments 2: Could you please provide additional information? Specifically, I'd like to know if this medium is suitable for the exclusive growth of G. pseudoferreum. Additionally, could you clarify any other relevant details about its preparation and usage?
Response 2: This medium was determined through our multiple researches and explorations as the suitable growth medium for G. pseudoferreum. The medium containing 10 g Tryptone, 5 g yeast powder, 20 g glucose Kermel, 10 g Sucrose Kermel, 2 g Mg2SO4, 2 g KH2PO4, 0.2 g Vitamin b1, 15 g Agar, 1,000 mL distilled water, and at pH 7.0 was prepared and sterilized at 121℃ for 30 minutes (Line-115).
Comments 3: Please provide manufacturer's details for all these aminoacids, sugars and vitamins
Response 3: Manufacturer's details has been provided in the manuscript (Line-115).
Comments 4: I'm not sure if 'pathogen cake' is a commonly used term. Could you please use 'mycelial growth' instead to describe the fungal culture that was cultivated on the PDA medium?
Response 4: Based on your valuable suggestion, we have changed " pathogen cake " to " Mycelial plugs "(Line-118).
Comments 5: Could you please provide information regarding the age of the rubber seedlings used in the experiment and specify the cultivar? This information is important for understanding the experimental conditions and interpreting the results accurately.
Response 5: We have added the relevant information in the manuscript (Line-125).
Comments 6: I'd like to clarify the region where the lesion area on the plants was measured. Could you please specify this information?
Response 6: The area covered by the mycelium of G. pseudoferreum is the measurement area.
Comments 7: The roots in the picture do not appear clearly defined. Please provide a clearer, higher-quality image to accurately depict the roots.
Response 7: Figures are available as supplementary material in the manuscript.
Comments 8: Scientific names italic problem
Response 8: All scientific names have been modified. I hope you can review it again

Reviewer 4 Report
Comments and Suggestions for Authors
Interesting paper identifying a beneficial bacteria that helps control pathogenic fungi of rubber trees. A few points need clarification and presentation needs to be tightened.
Throughout the paper and lines 16,17,21,23,35,36,37,61,65,274,324,515,557,572,573,580,587, figure 2 legend - italicize all organism scientific names
Line 51 - What are isolation ditches?
Line 75 - change to '...host, some of which do not cause visible harm...'
Line 76 - change 'endophytic' to 'plant'
Line 82 - change to 'Some beneficial endophytic...'
Line 136 and 358 - need to explain tridemorph
Line 164-165 - change 'thallus' to 'growth' or 'culture'
Line 399 - change to 4F
Line 403 - change to 4G
Comments on the Quality of English LanguageThe English for this paper need extensive editing. It really needs to be read over and edited by a native English speaker. I have marked some of the English/grammar problems below.
Line 45 - change 'China' to 'Chinese'
Line 47 - change to '...and causes large losses to the rubber industry [2].'
Line 48 - change 'was' to 'is'
Line 49 - change 'had the characteristic of' to 'has a'
Line 50-51 - change to '..cultural and chemical methods. Cultural controls include...'
Line 59 - delete 'Currently'
Line 63 - delete 'these'
Line 92 - delete 'In order'
Line 93 - delete 'an' and 'source'
Line 95 - change 'screening' to 'to screen'
Line 97 - change to '...seedlings, and [4]...'
Line 119 - sentence fragment
Line 126 - change to 'as described in section 2.1.'
Line 141 - sentence fragment
Line 187 - change to 'Each treatment was replicated..'
Line 267, 277, 287 - change to 'as described in section 2.X'
Line 314 - change 'referring to kit' to 'according to manufacturer's instructions'
Line 509 - change to '..production. Additionally,...'
Line 521-522 - Fix English
Line 531-533 - fix English
Line 550 - delete 'produced'
Author Response
Comments 1: Throughout the paper and lines 16,17,21,23,35,36,37,61,65,274,324,515,557,572,573,580,587, figure 2 legend - italicize all organism scientific names
Response 1: All scientific names have been modified. I hope you can review it again.
Comments 2: Line 51 - What are isolation ditches
Response 2: The isolation ditch refers to the creation of a deep trench between the diseased and healthy plots of land to prevent the spread of pathogens in the soil.
Comments 3: a), Line 75 - change to '...host, some of which do not cause visible harm...', b), Line 76 - change 'endophytic' to 'plant', c), Line 82 - change to 'Some beneficial endophytic...'
Response 3: We have revised the manuscript to include the following changes.
Comments 4: Line 136 and 358 - need to explain tridemorph
Response 4: Tridemorph is a pesticide for controlling red root rot disease
Comments 5: a), Line 164-165 - change 'thallus' to 'growth' or 'culture', b), Line 399 - change to 4F, c), Line 403 - change to 4G.
Response 5: We have revised the manuscript to include the following changes.
Comments 6: The English for this paper need extensive editing. It really needs to be read over and edited by a native English speaker. I have marked some of the English/grammar problems below.
Response 6: We have had the entire manuscript professionally proofread in English.

Round 2
Reviewer 3 Report
Comments and Suggestions for Authors
Dear Authors,
Thank you for making the changes to your manuscript. However, I wanted to note that you have identified your cultures as Phytophthora, but based on the pictures your provided, they appear to be Fusarium. I raised this concern previously, but it seems it was not addressed.
Additionally, there are instances where the introduction overlaps with the discussion, and the results are not discussed comprehensively. I recommend revisiting these sections to ensure clarity and thoroughness.
Thank you for your attention to these matters.

Comments on the Quality of English LanguagePlease see the attached document.
Author Response
Comments 1: Is it different from potato dextrose agar? Which flour?
Response 1: The Ganoderma pseudoferreum causing rubber tree red root disease was found to grow slowly in potato dextrose agar medium, while the use of flour instead of potato favoured the growth of G. pseudoferreum. The flour we used was wheat flour.
Comments 2: Have you confirmed their identity, particularly through molecular methods?
Response 2: Our strains were provided by professor Lifang Zou (Shanghai Collaborative Innovation Centre of Agri-Seeds/School of Agriculture and Biology, Shanghai Jiao Tong University, Shanghai 200240, P.R. China), and all strains have been authenticated, which can be referred to https://doi.org/10.3390/jof10020158.
Comments 3: While Photoshop can be a useful tool, I would like to suggest considering the use of Assess 2.0, a specialized image analysis software developed by the American Phytopathological Society (APS). Assess 2.0 is specifically designed for plant disease quantification and might offer more precise and standardized measurements of lesion size.
Response 3: Thank you for your suggestion, we recalculated the lesion size using the software (Assess 2.0) you recommended (Line-132).
Comments 4: Sima-Aldrich
Response 4: We have changed to Sigma-Aldrich
Comments 5: capsici? I previously raised a concern about the morphology of the image provided in the manuscript, which is identified as Phytophthora capsici. Upon review, the morphology in the image does not appear consistent with Phytophthora. It actually resembles characteristics more typical of some Fusarium species. Could you please provide clarification or further explanation regarding this discrepancy?
Response 5: We believe that the differences in strain morphology may be caused by issues such as photography, temperature and humidity, or contamination by miscellaneous bacteria. To avoid disputes over this issue, we have decided to remove the relevant images. We hope you will allow this.
Comments 6: This also does not belong here. In the "Discussion" you interpret results, significance of you rfindings,
Response 6: We have revised the "discussion" section (Line 507-517; 564-572).
Thank you for your consideration. I look forward to hearing from you.
